# Reduction of Biofouling of a Microfiltration Membrane Using Amide Functionalities—Hydrophilization without Changes in Morphology

**DOI:** 10.3390/polym12061379

**Published:** 2020-06-19

**Authors:** Daniel Breite, Marco Went, Andrea Prager, Mathias Kühnert, Agnes Schulze

**Affiliations:** Leibniz Institute of Surface Engineering (IOM), Permoserstr. 15, 04318 Leipzig, Germany; Daniel.Breite@iom-leipzig.de (D.B.); Marco.Went@iom-leipzig.de (M.W.); andrea.prager@iom-leipzig.de (A.P.); mathias.kuehnert@iom-leipzig.de (M.K.)

**Keywords:** polymer membrane, microfiltration, hydrophilization, amide coating

## Abstract

A major goal of membrane science is the improvement of the membrane performance and the reduction of fouling effects, which occur during most aqueous filtration applications. Increasing the surface hydrophilicity can improve the membrane performance (in case of aqueous media) and decelerates membrane fouling. In this study, a PES microfiltration membrane (14,600 L m^−2^ h^−1^ bar^−1^) was hydrophilized using a hydrophilic surface coating based on amide functionalities, converting the hydrophobic membrane surface (water contact angle, WCA: ~90°) into an extremely hydrophilic one (WCA: ~30°). The amide layer was created by first immobilizing piperazine to the membrane surface via electron beam irradiation. Subsequently, a reaction with 1,3,5-benzenetricarbonyl trichloride (TMC) was applied to generate an amide structure. The presented approach resulted in a hydrophilic membrane surface, while maintaining permeance of the membrane without pore blocking. All membranes were investigated regarding their permeance, porosity, average pore size, morphology (SEM), chemical composition (XPS), and wettability. Soxhlet extraction was carried out to demonstrate the stability of the applied coating. The improvement of the modified membranes was demonstrated using dead-end filtration of algae solutions. After three fouling cycles, about 60% of the initial permeance remain for the modified membranes, while only ~25% remain for the reference.

## 1. Introduction

One of the major challenges during membrane filtration is membrane fouling. The deposition of components from the filtered media onto the membrane naturally results in pore blockage and subsequently in the formation of a cake layer on top of the membrane [1,2,3]. The origin of those fouling effects can vary and depends on e.g., the membrane material (polymeric/ceramic), the membrane pore size (micro-/ultra-/nanofiltration/reverse osmosis), or the applied filtration mode (dead-end mode/cross-flow mode). As this publication is focused on microfiltration membranes, colloidal fouling, organic fouling, and biofouling have to be considered. Colloidal fouling is caused by the retention of e.g., particles due to the particles being larger than the membrane’s pore size. In contrast, organic fouling and biofouling are caused by interactions of the membrane with organic compounds like e.g., proteins or microorganisms like algae. While colloidal fouling represents a desired effect of the membrane filtration (retention according to the size of an object), organic fouling and biofouling are undesired and have a negative impact on a membrane’s filtration performance and lifetime. To reduce the influence of organic fouling and biofouling, it is necessary to decrease the attractive interactions between the membrane and the foulants solved in the filtered media. The interactions between membranes and foulants are mainly based on hydrophobic and electrostatic interactions. The reduction of these interactions can be realized by modifying the membrane surface via changes in e.g., the surface charge, morphology, or surface hydrophilicity [4,5,6,7,8,9,10,11,12,13,14,15,16,17,18,19,20,21].

While different methods have been successfully applied to alter the surface morphology or surface charge of a membrane, changing the surface hydrophilicity is the far more prominent approach. The increase of the surface hydrophilicity results in the formation of a thin water film on top of the polymer membrane. This water film helps to prevent hydrophobic interactions between the polymer of the membrane and hydrophobic components solved in the filtered solution [4]. Examples for membrane hydrophilization methods are the application of (UV-initiated) grafting reactions [5,6,7,10,12,22,23], plasma treatments [8,9,24,25,26,27,28,29,30,31], or surface modifications initiated by electron beam irradiation [32,33,34]. UV-based methods are only capable to modify the top layer of a membrane (due to the restricted interpenetration depth of UV light) and plasma treatments are known to be unstable over time (due to hydrophobic recovery of the hydrophilized polymer chains). In comparison, electron beam irradiation enables to permanently modify the top side of a membrane as well as its porous structure [32,33,34]. Furthermore, electron beam irradiation can be applied in a roll-to-roll process similar to UV irradiation, and thus, enables easy upscaling.

Contrary to a hydrophobic polymer membrane, hydrophilic ones do not require an additional surface hydrophilization. For example polyamide (PA) membranes show very good performance in industrial nanofiltration and reverse osmosis applications [35,36]. This kind of membrane is prepared as a composite membrane with a thin PA layer (by interfacial polymerization) on top of a porous membrane made from a different (hydrophobic) polymer. Here, the characteristics of PA lead to a hydrophilic membrane surface. However, as it is the purpose of these applications, the resulting membranes have a dense PA layer. The application of the interfacial polymerization process needs e.g., an ultrafiltraftion membrane as a base material. The polymerization takes place at the interface between an aqueous phase containing the amine component and an organic phase containing the acid chloride. This interface is usually located on the top side of the membrane, which results in the formation of the PA layer on top of the base membrane. More open pore structures like e.g., microfiltration membranes can also be used as a base membrane, but thick PA layers are necessary to generate a stable composite system. Furthermore, the original permeance of the microfiltration membrane is “lost”. However, since PA has already demonstrated to be an interesting hydrophilic polymer for membrane applications, it could be an interesting approach to use amides in the context of a surface modification. As discussed before, interfacial polymerization is not suited as a hydrophilization method for microfiltration membranes. However, the basic idea of the interfacial polymerization concept can be applied using the electron beam-based immobilization process presented in this study. By first immobilizing the amine component via an electron beam-induced grafting-to reaction and a subsequent reaction with the acid chloride component, a thin amide layer is generated. This amide layer is present on the top side of the membrane as well as in the membrane pores improving the overall hydrophilicity of the membrane. At the same time, the permeance of the membrane is not significantly decreased. The hydrophilic properties of PA have already been reported in the literature for e.g., membrane prepared from PA [37]. In this study, the hydrophilic characteristics of PA are combined with the mechanical stability of a PES microfiltration membrane. The presented innovative approach was inspired by PA interfacial polymerization but transformed to a new surface modification technique.

To confirm the effect of membrane modification the resulting fouling behavior needs to be investigated. Many studies found in the literature are using artificial solutions of e.g., proteins like BSA or natural organic matter (NOM). However, the composition of e.g., surface water can vary drastically, so other foulant should also be considered. In this study, filtrations were carried out using algae solutions as a model system representing a major component of e.g., surface water. While the effect of algae bloom is reported in literature [38], publications describing the impact of the actual algae solutions are rarely found. Hung and Liu presented a study on a microfiltration process for the separation of green algae [39], while Qu et al. focused on the fouling caused by extracellular organic matter released from algae [40]. Wu et al. demonstrated the suppression of algae fouling by creating a polyamide layer via interfacial polymerization on top of an ultrafiltration base membrane. The polyamide of this thin film composite membranes was composed of alginate and 1,3,5-benzenetricarbonyl trichloride (TMC) [41].

## 2. Materials and Methods

### 2.1. Reagents and Materials

Non-solvent induced phase separation (NIPS) was used to prepare polyether sulfone (PES) flat sheet membranes. Purchased chemicals: 1-methyl-2-pyrrolidone (NMP, Sigma Aldrich, St. Louis, MO, USA), 1,3,5-benzenetricarbonyl trichloride (TMC, Sigma Aldrich, St. Louis, MO, USA), *n*-hexane (Merck, Darmstadt, Germany), piperazine (PIP, Sigma Aldrich, St. Louis, MO, USA), polyether sulfone (PES, Ultrason E2010, BASF, Ludwigshafen, Germany), polyethylene glycol (PEG, 400 g mol^−1^, Acros Organics, part of Thermo Fisher Scientific, Geel, Belgium). 

The algae solutions were prepared using *chlorella vulgaris* provided by UTC Prague, Department of Biotechnology and were grown according to the description given in Section 2.5. The following chemicals purchased from VWR (Radnor, USA) were used for the culture medium: urea, potassium dihydrogen phosphate, magnesium sulfate heptahydrate, ethylenediaminetetraacetic acid (EDTA), calcium chloride, boric acid, copper sulfate pentahydrate, manganese chloride pentahydrate, cobalt sulfate heptahydrate, zinc sulfate heptahydrate, ammonium molybdate tetrahydrate, ammonium vanadate.

All chemicals were used as received. Ultrapure water was taken from a MilliQ-System (Merck Millipore, Billerica, MA, USA). 

### 2.2. PES Membrane Preparation

PES membranes were prepared by NIPS. A solution containing 14 wt.-% PES, 65 wt.-% PEG, and 21 wt.-% NMP was casted (200 µm gap, ZWA 2121 Wasag Applicator, Zehntner Testing Instruments, Sissach, Switzerland) on a glass plate and kept in humidified air for 5 min. The precipitation was performed in a water bath (~10 °C, 10 min), followed by washing the membranes with pure water three times for 30 min, respectively. 

### 2.3. Membrane Modification

Three different types of amide modifications were applied to the PES membranes. For the first (PES-PIP-TMC) and second approach (PES-PIP-TMC-PIP), piperazine (PIP) was immobilized to the membrane in a first step via electron beam immobilization. The modification was carried out using a self-made electron accelerator in nitrogen atmosphere (O_2_ quantities <10 ppm) [33]. Membrane samples were immersed (30 min) into an aqueous solution of PIP (2 wt.-%) and irradiated using a dose of 200 kGy. The membranes were washed with pure water three times for 30 min, respectively, and were dried, subsequently. The membranes were then mounted in a stainless steel pressure filter holder (16249, Sartorius, Goettingen, Germany) to apply the second modification step. 1,3,5-benzenetricarbonyl trichloride (TMC, 100 mL, 0.2 wt.-% in *n*-hexane) was passed through the membranes at a pressure of 0.5 bar. The membranes were dried to remove the *n*-hexane and were washed, subsequently, with pure water three times for 30 min, respectively. In the case of the PES-PIP-TMC-PIP membranes, a third reaction step was applied. The membranes were once more immersed in a PIP solution (0.2 wt.-% in water) for 30 min, followed by washing the samples with pure water three times for 30 min, respectively.

For comparison, PES membranes were also modified with a dense polyamide layer (PES-PA) following the classical approach of an interfacial polymerization reaction. Therefore, a sample of ~10 × 10 cm was immersed in a PIP solution (0.2 wt.-% in water [42]) and subsequently mounted in a stainless steel frame (photographic images of the metal frame are shown in Figure 1). Excess water on the top side of the membrane was removed and TMC solution (0.2 wt.-% in *n*-hexane) was added to top side of the metal frame. Due to the metal frame, the polymerization reaction could only take place at the top side of the PES membrane, where the interface between the organic and inorganic phases is located. The TMC solution was removed after 2 min and the membrane was released from the metal frame. The membranes were dried to remove the *n*-hexane and subsequently washed with pure water three times for 30 min, respectively.

All membranes were stored in water at room temperature until further usage. An overview of the applied modification steps and solutions in presented in Table 1. 

### 2.4. Membrane Characterization

All membranes were investigated via scanning electron microscopy (SEM), measurement of water permeance, and mercury porosimetry regarding the membrane morphology. X-ray photoelectron spectroscopy (XPS), and water contact angle (WCA) measurements were carried out to analyze the chemical composition and surface hydrophilicity of the membrane surfaces, respectively. 

SEM (Ultra 55 SEM, Carl Zeiss Ltd., Goettingen, Germany) was used to investigate the membrane morphology. A chromium coating (30 nm, Z400 sputter system, Leybold, Hanau, Germany) was used to prevent charging of the samples.

Water permeance was calculated based on filtration experiments using a stainless steel pressure filter holder (16249, Sartorius, Goettingen, Germany) for dead end filtration. 100 mL of deionized water were filtered through the membrane (active area: 17.3 cm^2^) at 1 bar, and the time of flow-through was recorded. Values of at least three different (independently prepared) samples were averaged. Pure water permeance *J* was calculated following Equation (1).
(1)$$J=\frac{V}{t\cdot A\cdot p}$$

*V* is the volume of water, *t* the time of flow-through, *A* the active area, and *p* the applied pressure. 

Porosity and pore size distribution were measured with a mercury porosimeter (PoreMaster 30, Quantachrome Instruments, Odelzhausen, Germany). Values of at least three different samples were averaged.

X-ray photoelectron spectroscopy (XPS, Kratos Axis Ultra, Kratos Analytical Ltd., Manchester, UK) was used to analyze the chemical composition of the membrane surface. Values of at least three different (independently prepared) samples were averaged.

Surface wettability was measured using a static water contact angle measurements system (DSA 30E, Krüss, Hamburg, Germany) and the sessile drop method. Values of at least three different (independently prepared) samples were averaged. To also investigate the very hydrophilic surfaces of the modified membranes, movies were recorded of the drops deposited on the membrane surfaces. The initial contact angle (after stabilizing of the drop, before subsiding in the membrane) was measured using a single frame to gain data comparable to the hydrophobic reference membranes. 

The stability of the modified membranes was investigated via Soxhlet extraction using water as solvent. Membrane samples were extracted for 5 d at 100 °C followed by membrane characterization (permeance, water contact angle, XPS) as described above. 

### 2.5. Preparation of Fouling Solutions

Algae solutions were prepared by growing *chlorella vulgaris* in a self-build reactor system (150 mL, 30 °C, light source of 445 nm, 2.5 L min^−1^ air bubbling) for 7 d [43]. The culture medium was composed of urea (1100 mg L^−1^), potassium dihydrogen phosphate (238 mg L^−1^), magnesium sulfate heptahydrate (204 mg L^−1^), EDTA (40 mg L^−1^), calcium chloride (88 mg L^−1^), boric acid (0.832 mg L^−1^), copper sulfate pentahydrate (0.946 mg L^−1^), manganese chloride pentahydrate (3.294 mg L^−1^), cobalt sulfate heptahydrate (0.616 mg L^−1^), zinc sulfate heptahydrate (2.678 mg L^−1^), ammonium molybdate tetrahydrate (0.172 mg L^−1^), ammonium vanadate (0.014 mg L^−1^). The pH value of the solution was adjusted to 7. After 7 d of growth, the algae solutions were removed from the reactor and stored at room temperature in a stirred vessel. Algae solutions should be used within 5 d after preparation. To gain comparable algae concentrations for all fouling experiments, the algae solutions were diluted using pure water. The light absorption at 680 nm (200 µL, 48-well plate, Infinite M200, Tecan, Männedorf, Switzerland) was controlled using a calibration curve and the total organic carbon (TOC, liquiTOC II, Elementar Analysensysteme GmbH, Hanau, Germany) was measured. Algae solutions used in this study had a light absorption of 0.07 at 680 nm and a TOC of 25 ± 1 mg L^−1^.

### 2.6. Fouling Test

Filtration experiments with algae solutions were performed by dead-end filtration using a 50 mL stirred cell (Amicon, Merck Millipore, Billerica, MA, USA). The membrane sample (active area: 15.9 cm^2^) was mounted into the stirring cell and a volume of 50 mL water was passed through the membrane at 1 bar, while recording the time of flow-through to determine the initial water permeance. Afterwards, 50 mL of algae solution were passed through the membrane and the time of flow-through was measured for every 10 mL step. Then, the membrane was turned upside down and 50 mL of pure water were passed through the membrane for backwashing at 1 bar. After remounting the membrane in its original direction, 50 mL of water were passed through the membrane to determine the clean water permeance. The complete cycle of fouling, backwash, and water permeance measurement was repeated three times. Values of at least three different (independently prepared) samples were averaged.

## 3. Results and Discussion

### 3.1. Membrane Modification with Amide Fuctionalities

Three types of amide modifications were applied to the PES base membrane. To obtain the PES-PIP-TMC membrane, PIP was immobilized on the membrane surface via electron beam immobilization (Figure 2). Due to the electron beam irradiation of the PES membrane immersed in an aqueous solution of PIP, radical species are formed. Subsequent recombination reactions lead to a covalent binding of PIP molecules to the membrane surface. After removing and drying of the modified membranes, TMC solved in *n*-hexane was flushed through the membranes. Thus, the acid chloride group of TMC reacts with the amine function of PIP forming an amide group. After removal of *n*-hexane and an additional washing step, the PES-PIP-TMC membrane was formed. It was considered to just immerse the PES-PIP membrane in a TMC solution. However, the resulting membranes were inhomogeneous and non-reproducible. Filtration of the TMC solution resulted in much more reliable membrane modifications. It can be assumed that active filtration led to a better contact between TMC and the amine groups, resulting in more homogenous modification.

The preparation of the PES-PIP-TMC-PIP membrane follows the same procedure described above, adding a third reaction step. By immersing a PES-PIP-TMC once more in an aqueous PIP solution, the remaining acid/acid chloride groups (the acid chloride was most probably hydrolyzed due the washing steps using water) of the former TMC molecule can thus react with PIP from the solution forming another amide moiety. 

For comparison, a membrane (PES-PA) was also prepared following the classical approach of an interfacial polymerization reaction resulting in a dense PA on top of the PES base membrane. Therefore, a PES membrane was immersed in an aqueous PIP solution and subsequently mounted in a stainless steel frame. Excess water on the top side of the membrane was removed and a solution of TMC in *n*-hexane was added to top side of the metal frame. The amide formation took place as described above. However, due to the PIP molecules diffusing from the solution inside the porous membrane into the organic phase [35], the reaction continues forming a dense polyamide network. In contrast, in case of the new electron beam-based approach the only PIP molecules present for reaction were previously immobilized on the membrane surface. 

The reference membrane as well as the three modified membranes were thoroughly characterized using SEM to investigate morphological changes, permeance measurements to see changes in the membranes performance, mercury porosimetry to detect possible pore narrowing, water contact angle (WCA) measurements to demonstrate changes in the membranes surface hydrophilicity, as well as XPS to observe changes in the chemical composition of the membrane surfaces. 

A summary of all investigated membrane characteristics is presented in Table 2. A comparison of SEM images of the membranes top sides (active side) is given in Figure 3. The SEM images of the reference membrane show the open pore structure of this PES microfiltration membrane. No visible changes were observed for the membranes modified using electron beam-based approaches (PES-PIP-TMC and PES-PIP-TMC-PIP). The modification layer of these membranes is very thin and cannot be detected using SEM, but the presence of the amide functionalities was proven using other methods such as e.g., XPS. A dense top layer is observed in the case of the PES-PA membrane. Due to the polymeric amide network formed by the interfacial polymerization, the pore structure of the PES base membrane is mostly blocked. 

The results of the SEM analysis are in accordance with the measured permeance values (Table 2). A permeance of 14,600 L h^−1^ m^−2^ bar^−1^ was determined for the unmodified reference membrane. Slightly decreased values were achieved for the PES-PIP-TMC and PES-PIP-TMC-PIP membranes. The slight decrease could originate from pore narrowing from the modification (not visible in SEM) or more likely from individual differences of the self-made membranes. In contrast, the PES-PA membrane has a significantly reduced permeance of 650 L h^−1^ m^−2^ bar^−1^, as expected considering the dense PA layer on top of the base membrane. The porosity and average pore size of the reference membrane were determined to be 72% and 0.65 µm, respectively. The porosity of all modified membranes were found to be ~53%. This reduction could be attributed to the membrane modification. Nevertheless, as self-made membranes were used in this study, a variation in the membrane batch cannot be completely excluded. The average pore size is also reduced for the modified membranes, where the smallest pore size of 0.46 µm was observed in the case of the PES-PA membrane. 

One of the key characteristics of this study was the surface hydrophilicity determined via water contact angle (WCA) analysis. The reference membrane was only composed of PES and thus, exhibits a hydrophobic surface with a WCA of 89° (ATR-IR measurements confirmed the absence of PEG from the membrane preparation. These results are not discussed here, as this type of membrane was used in prior publications [14,44,45].) The dense PA layer of the PES-PA membrane resulted in a significantly more hydrophilic surface with a WCA of 59°. However, as discussed before, this membrane can no longer be used as a microfiltration membrane. This is not the case for the PES-PIP-TMC and PES-PIP-TMC-PIP membranes. The amid functionalization of these membranes resulted in very low WCA values of 37° and 30°, respectively. These membranes are very hydrophilic, while still usable for microfiltration application. The electron beam-based modification are chemically similar to the polyamide layer of the PES-PA membrane. However, the very hydrophilic surface characteristic results from the combination of the open porous structure of the membrane and the hydrophilic PA layer. The higher roughness, and thus higher surface area, of the open pore structure compared to the smooth PA layer of the PES-PA membrane leads to a decrease of the WCA according to Wenzel’s law [46].

XPS analysis was carried out to confirm the chemical composition of the different membrane surfaces. The reference membranes was composed of 74% carbon, 20% oxygen, and 5% sulfur. This is close to the theoretical values expected for pure PES. The application of a polyamide layer (PES-PA membrane) changed the composition measured at the membrane surface. 71% carbon, 19% oxygen, 1% sulfur, and 8% nitrogen were detected. The presence of high amounts nitrogen clearly proves the successful formation of the dense PA layer. The small amount of sulfur originates from the PES base membrane, which can still be detected below the PA layer. In case of the PES-PIP-TMC and PES-PIP-TMC-PIP, the applied amide layer is much thinner compared to the PA layer of the PES-PA membrane. Thus, only 1–2% nitrogen are detected to confirm the presence of the amide functionalization. 

### 3.2. Algae Fouling

Filtration test were carried out for the reference membrane, the PES-PIP-TMC membrane, as well as for the PES-PIP-TMC-PIP membrane to investigate the membrane performance of the modified membranes. No filtration tests were performed for the PES-PA membrane as this membrane has a dense top layer and was only included in this study for comparison of the membrane characteristics as discussed above. 

The filtrations were carried out using algae solutions. This test system was chosen because algae are one major component of surface water [40] and because fouling with algae was rarely investigated in the literature. Algae solution was grown for one week prior to usage for filtration experiments. Thus, the algae solutions contained algae as well as metabolites formed during the growth period. 

Filtration experiments were carried out in dead-end mode under constant pressure. After the determination of the initial water permeance, a volume of a defined algae solution was filtered through the membranes. The algae (size of ~3–5 µm) as well as the other components present in the algae solution led to membrane fouling by blocking the pore structure. The thus fouled membranes were backwashed to remove the loosely bound fouling components, followed by determining the pure water permeance. This cycle was repeated three times. Figure 4 presents the course of the fouling experiments for the reference membrane, the PES-PIP-TMC membrane, as well as the PES-PIP-TMC-PIP membrane. 

The overall course of all three membranes follows a similar pattern. The permeance decreases during the algae filtration steps and recovers during the subsequent backwash and pure water filtration steps. However, a significant difference can be observed for the reference membrane compared with both modified membranes. The decrease in the permeance is much more pronounced during the algae filtration step regarding the reference membrane. After algae filtration, the permeance of the reference membrane was found to be about 10% of the initial permeance. In case of both modified membranes a permeance of ~40% remains. After the first backwash a pure water permeance of about 40% was determined for the reference membrane, while the modified membranes recovered to ~80%. It can be assumed that the majority of the pore blocking originates from algae deposited on the membrane. As the spherical algae used in this study are bigger than the membrane pores, they cannot pass the membrane and are retained due to size exclusion. However, the algae solutions also contain other components from the algae metabolism such as e.g., proteins. These component can foul the membranes due to hydrophobic interaction or other attractive interactions. The increased hydrophilicity of the modified membrane leads to the formation of an additional water film on top of the membrane preventing such interactions. Thus, the overall permeance decrease of the modified membranes is less affected by other components from the algae solution compared with the reference membrane. 

The repetition of the fouling-cleaning-cycles leads to an overall decrease in the pure water permeance measured at the end of each cycle. However, after three cycles about 60% of the initial permeance remain for the modified membranes, while the reference membrane has ~25% permeance compared to the initial pure water measurement. Thus, even in long-term usage the modified membranes have a significant advantage compared to the reference membrane. 

SEM investigations (Figure 5) were carried out for the membranes following the fouling experiments described above. The top side of reference membrane, the PES-PIP-TMC membrane, as well as the PES-PIP-TMC-PIP membrane showed a small number of algae, which remained after the final backflush and pure water measurement. The algae cells appear to have shrunk because the sample preparation and measurements were carried out in vacuum. However, the morphology of the cells is secondary and will not affect number of cells, which can be observed. 

A significant difference can be observed regarding the reference membrane. This membrane has a number of areas, which are covered by organic fouling (not algae) originating from the algae solutions. This organic fouling is most probably composed of proteins and other organic metabolites from the algae growth. The composition of comparable algae solutions was already reported in literature [41]. Due to the hydrophobic character of the reference membrane, fouling occurs and irreversibly blocks the membrane pores. The hydrophilic surface of the modified membranes prevents most of these interactions, significantly reducing the organic fouling of these membranes. 

No significant differences were observed comparing the PES-PIP-TMC membrane and the PES-PIP-TMC-PIP membrane in this part of the study. However, as some differences occurred in the latter part of this work, both membrane were discussed here as well. 

### 3.3. Stability of the Amide Modification

To make the modifications presented in this study applicable to an actual microfiltration application, it is necessary to demonstrate their long-term stability. Because long-term filtration experiments cannot be conducted in dead-end mode, the stability of the membrane modifications was investigated using Soxhlet extractions. By extracting the membrane samples for 5 d using hot water, all soluble and not permanently immobilized parts of the modification layer should be removed. Thus, a long-term usage using water at ambient temperature can be simulated. Water permeance measurements, XPS, as well as filtration experiments with algae solutions were carried out to characterize the membrane samples after the stability test. 

A comparison of the characteristics of the modified membranes before and after Soxhlet extraction is given in Table 3. The permeance values after Soxhlet extraction are slightly increased compared to the initial values, but the changes are close to or within in the calculated errors. The same is true for the XPS data before and after stability testing. Only slight variations occur and no significant differences are observed. Thus, it can be concluded that the amide functionalization is still present even after a harsh treatment like the long-time exposition to hot water. WCA measurements were not conducted, as the input of thermal energy can lead to rearrangements of the polymer chains also known as hydrophobic recovery. This effect decreases the number of hydrophilic groups available at the membrane surface after drying, and thus, falsifies the WCA measurements.

To investigate the membrane performance after Soxhlet extraction, filtration experiment were carried out using algae solutions and the same procedure describe above. A comparison of the course of the filtration experiments before and after the stability test is presented in Figure 6. 

It can be observed that the decline in permeance during the algae filtration steps is more pronounced for the samples treated with hot water. However, the ability to regenerate the permeance (~60% after the first filtration cycle) of the membranes is still present. It can be concluded that the amide modification is mostly stable even after Soxhlet extraction. A bigger deviation between the membranes before and after stability testing can be observed at the end of third filtration cycle. While the remaining pure water permeance of the membranes before the stability test was at ~60% (reference membrane: ~25%), the final permeance of the modified membranes after Soxhlet extraction was found to be 40% (PES-PIP-TMC) and 45% (PES-PIP-TMC-PIP). This decrease could be attributed e.g., to a partial loss of not covalently bound modification layers. Furthermore, the final value of the PES-PIP-TMC-PIP membrane was slightly better than that of the PES-PIP-TMC membrane, which might indicate a better stability of membranes treated with the additional modification step. Nevertheless, the pure water permeance after three filtration cycles is still better for the modified membranes (40%/45%) compared to the value determined for the reference membrane (~25%) without a prior hot water treatment. Thus, it can be concluded that the applied modifications increased the membrane performance also under long-term conditions. 

## 4. Conclusions

This publication presented a new modification approach to hydrophilize PES microfiltration membranes (14,600 L m^−2^ h^−1^ bar^−1^) with a thin amide layer. As a result, the WCA was reduced from 89° to 30°. In contrast to a classical interfacial polymerization for preparation of composite membranes, this innovative electron beam-based approach enables the introduction of amide functionalities on the surface of a porous microfiltration membrane without blocking its porous structure. Thus, this method enables the creation of a very hydrophilic surface while maintaining the open pore structure. 

Furthermore, the improved membrane performance was demonstrated in fouling experiments using algae solutions as a major component of e.g., surface water. The modified membranes retained about 60% of their original pure water permeance, whereas the permeance of a reference membrane was reduced to 25%. Finally, the stability of the applied surface modifications was confirmed using a Soxhlet extraction method to simulate a long-term usage under ambient conditions. 

## Figures and Tables

**Figure 1 polymers-12-01379-f001:**
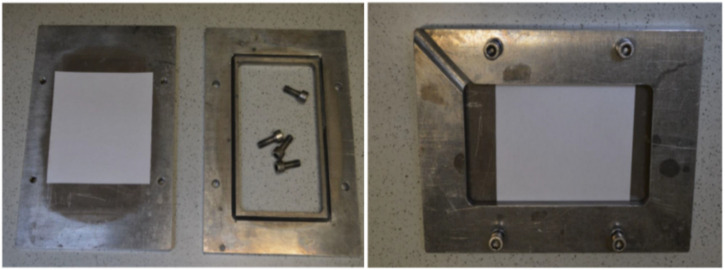
Metal frame for the preparation of the PES-PA membrane: disassembled (**left**) and assembled (**right**). The membrane was depicted by a sheet of paper.

**Figure 2 polymers-12-01379-f002:**
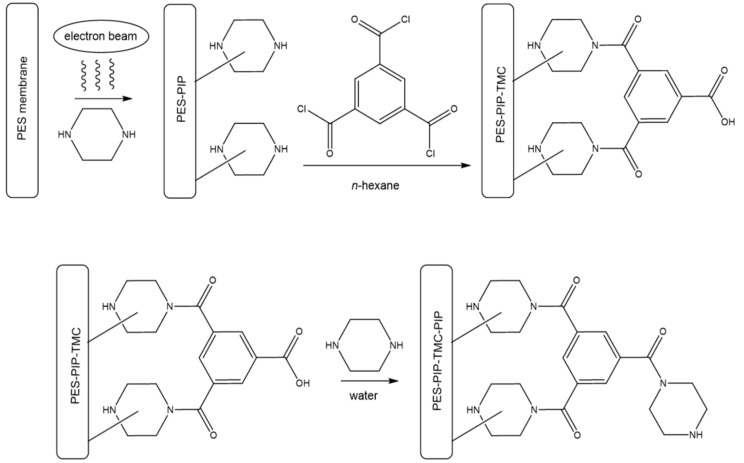
Preparation of the PES-PIP-TMC and PES-PIP-TMC-PIP membranes.

**Figure 3 polymers-12-01379-f003:**
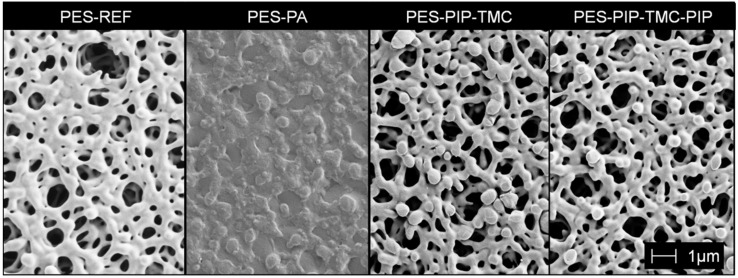
SEM images (top side) of the reference membrane and the three modified membranes.

**Figure 4 polymers-12-01379-f004:**
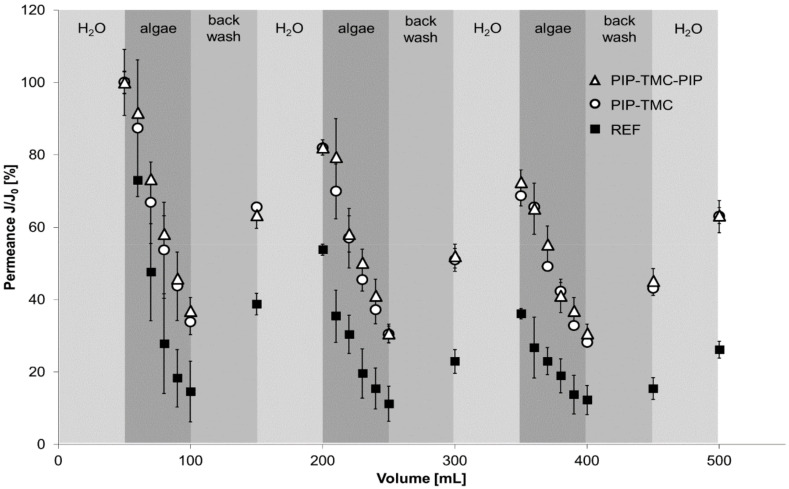
Filtration experiments using algae solution with the reference membrane (filled squares), the PES-PIP-TMC membrane (open circles), and the PES-PIP-TMC-PIP membrane (open triangles).

**Figure 5 polymers-12-01379-f005:**
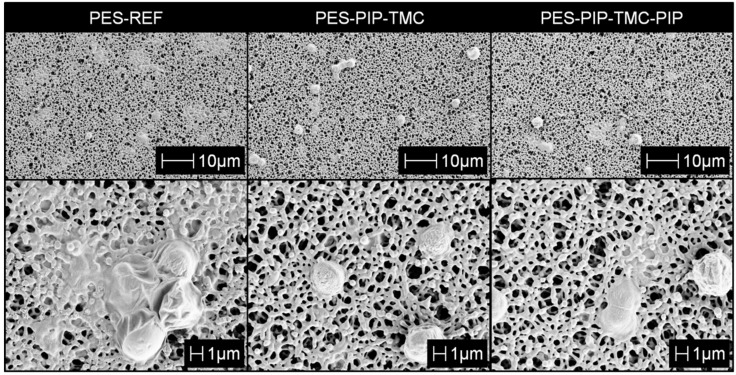
SEM images (top side) of the different membranes after filtration of algae solution at different magnifications: 1000-fold (**top**) and 5000-fold (**bottom**).

**Figure 6 polymers-12-01379-f006:**
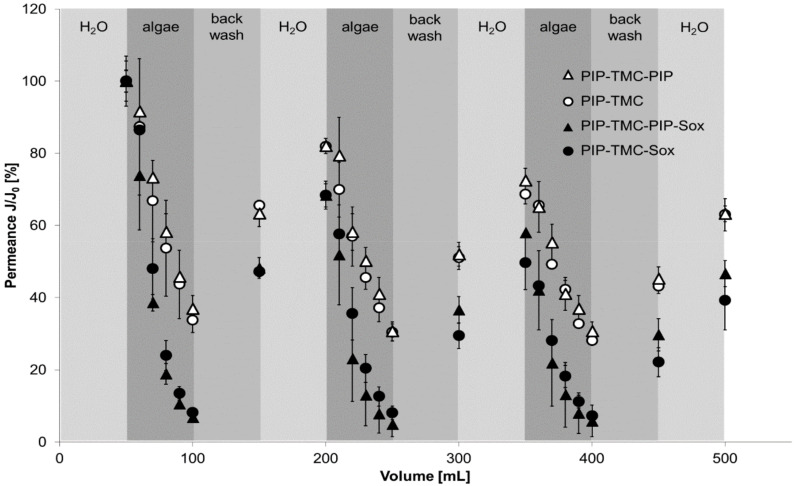
Filtration experiments using algae solution with the PES-PIP-TMC membrane before (filled circles) and after Soxhlet extraction fore (open circles), and the PES-PIP-TMC-PIP membrane before (filled triangles) and after Soxhlet extraction fore (open triangles).

**Table 1 polymers-12-01379-t001:** Membrane modifications.

Approach	PIP Conc. (30 min)	Electron Beam Dose	TMC Conc.	TMC Application Via	2nd PIP Conc. (30 min)
	[wt.-%]	[kGy]	[wt.-%]		[wt.-%]
PES-PA	0.2	-	0.2	contact on top side for 2 min	-
PES-PIP-TMC	2	200	0.2	filtration at 0.5 bar	-
PES-PIP-TMC-PIP	2	200	0.2	filtration at 0.5 bar	0.2

**Table 2 polymers-12-01379-t002:** Membrane characterization.

Membrane	Permeance	Porosity	Average Pore Size	Water Contact Angle	Chemical Composition (XPS) [rel. atom-%]			
	[L (m^2^ h bar)^−1^]	[%]	[µm]	[°]	C	O	N	S
REF	14,600 ± 700	72 ± 2	0.65 ± 0.04	89 ± 5	74 ± 2	20 ± 2	-	5 ± 1
PES-PA	650 ± 150	53 ± 5	0.46 ± 0.05	59 ± 5	71 ± 2	19± 1	8 ± 2	1 ± 1
PES-PIP-TMC	11,500 ± 300	53 ± 1	0.58 ± 0.02	37 ± 5	74 ± 1	21 ± 2	1 ± 1	4 ± 1
PES-PIP-TMC-PIP	12,300 ± 400	52 ± 1	0.55 ± 0.02	30 ± 3	73 ± 1	21 ± 2	2 ± 1	4 ± 1

**Table 3 polymers-12-01379-t003:** Membrane characterization before and after Soxhlet extraction.

Membrane		Permeance	Chemical Composition (XPS) [rel. atom-%]			
		[L (m^2^ h bar)^−1^]	C	O	N	S
PES-PIP-TMC	before Soxhlet	11,500 ± 300	74 ± 1	21 ± 2	1 ± 1	4 ± 1
	after Soxhlet	13,100 ± 900	74 ± 3	21 ± 1	1 ± 1	5 ± 2
PES-PIP-TMC-PIP	before Soxhlet	12,300 ± 400	73 ± 1	21 ± 2	2 ± 1	4 ± 1
	after Soxhlet	12,600 ± 700	73 ± 2	20 ± 3	2 ± 1	6 ± 2

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
