# Peer review of "Reduction of Biofouling of a Microfiltration Membrane Using Amide Functionalities—Hydrophilization without Changes in Morphology"

_polymers, 2020, doi:10.3390/polym12061379_

Round 1

Reviewer 1 Report

This study adapted an interfacial polymerization commonly used to prepare nanofiltration membranes to a surface modification method, which is very interesting. This manuscript is acceptable after addressing the following comments.

P2, line93. A typo in the sentence.

P3, line132. For the case of PES-PIP-TMC-PIP, the third step was applied to the membrane with or without water washing? TMC can react with water.

P3, line138. In the comparison study, the concentration of the PIP used is 0.2 wt%, rather than 2 wt%. It would be better to use the same concentration as the previous study.

It was mentioned in the abstract and introduction part that surface charge also influences the fouling of a membrane. So, how about the surface charge of these modified membranes?

What about the water contact angle of the modified membrane after the stability test?

Author Response

Thank you very much for considering our manuscript for publication in Polymers. We appreciate your thorough examination of the manuscript, and we carefully revised the manuscript for obtaining now a better quality of the paper. Therefore, we would like to resubmit our manuscript and hope we were able eliminate all mistakes and misunderstandings according to your suggestions.

We responded to the different review comments as follows:

This study adapted an interfacial polymerization commonly used to prepare nanofiltration membranes to a surface modification method, which is very interesting. This manuscript is acceptable after addressing the following comments.

P2, line93. A typo in the sentence.

The typo was corrected.

P3, line132. For the case of PES-PIP-TMC-PIP, the third step was applied to the membrane with or without water washing? TMC can react with water.

PES-PIP-TMC and PES-PIP-TMC-PIP were handled the same until the end of the second step, including washing with water at the end. Thus, remaining (unbound) TMC will be removed before the third step (PIP). We are aware, that TMC reacts with water. Therefore, the bound TMC groups were discussed as acid/acid chloride groups in the discussion part (line 230).

P3, line138. In the comparison study, the concentration of the PIP used is 0.2 wt%, rather than 2 wt%. It would be better to use the same concentration as the previous study.

The PIP concentration of 0.2 wt.-% were chosen according to what was found for interfacial polymerization reactions in the literature. A respective citation was added in the experimental part (line 140). The higher concentration of PIP for the electron beam irradiation step stemmed from our experience with that system, where higher concentrations (~0.5-2 wt.-%) give a better immobilization of the functional molecule. All other PIP concentrations (PES-PA and third step of PES-PIP-TMC-PIP) remained at the original concentration of 0.2 wt.-%.

It was mentioned in the abstract and introduction part that surface charge also influences the fouling of a membrane. So, how about the surface charge of these modified membranes?

The surface charge of the reference and modified membranes were investigated via zeta potential analysis, but no noteworthy changes were detected. All membranes had a negative potential at the pH value used in the filtration experiments. The zeta potential of the algae used in this study is also known to be negative at the used pH value. Thus, the electrostatic interactions (here: electrostatic repulsion) remain nearly the same before and after membrane modification. Accordingly, the observed effects should be attributed to the improvements in wettability of the membranes.

What about the water contact angle of the modified membrane after the stability test?

Soxhlet extraction was applied to test the stability of the membranes. While this technique is very versatile, it has one major drawback, which is the input of thermal energy in the system. The thermal energy can lead to rearrangements of the polymer chains also known as hydrophobic recovery. This effect decreases the number of hydrophilic groups available at the membrane surface after drying, and thus, falsifies the water contact angle measurements. For this reason, we usually do not conduct water contact angle after Soxhlet extraction. The XPS data demonstrate well that the surface functionalization is still present.

Reviewer 2 Report

Reviewer 2:Comments and Suggestions for Authors Breite

The manuscript ID polymers-780848 and titled as “Reduction of Biofouling of a Microfiltration Membrane using Amide Functionalities – Hydrophilization without Changes in Morphology” was reviewed. It is interesting paper. I accept the paper after some revision as indicated below.

  1. Abstract should be more quantitative. Some technical part should insert.
  2. The digital image of prepared membrane should insert.
  3. Introduction needs some more recent papers by 2019 and 2020. Introduce Carbohydrate Polymers, 211(2019)181-194, regarding UV (Check Line 53-57). Novelty of the research should be mention at the end of introduction. Recently reported similar work needs more comparison with present work.
  4. The equation 1 should be separated from characterization part. Modify and check it.
  5. Line 251-253 needs proper references.
  6. Line 276-278, WCA of amide functionalization for membrane should be referred.
  7. Line 310-312 should be rechecked.
  8. Conclusion should be technical.
  9. No need of the supplementary, you can insert in this part in text. Check and modify it.

Author Response

Thank you very much for considering our manuscript for publication in Polymers. We appreciate your thorough examination of the manuscript, and we carefully revised the manuscript for obtaining now a better quality of the paper. Therefore, we would like to resubmit our manuscript and hope we were able eliminate all mistakes and misunderstandings according to your suggestions.

We responded to the different review comments as follows:

The manuscript ID polymers-780848 and titled as “Reduction of Biofouling of a Microfiltration Membrane using Amide Functionalities – Hydrophilization without Changes in Morphology” was reviewed. It is interesting paper. I accept the paper after some revision as indicated below.

    Abstract should be more quantitative. Some technical part should insert.

Some quantitative data was included in the abstract as requested by reviewers 2 and 3. However, due to the limited number of words for the abstract, another sentence was removed.

    The digital image of prepared membrane should insert.

The authors do not understand what kind of image is requested by the reviewer. A SEM image of the reference membrane is shown in figure 3 (former figure 2). What should be shown on the requested image (SEM or photographic picture)?

    Introduction needs some more recent papers by 2019 and 2020. Introduce Carbohydrate Polymers, 211(2019)181-194, regarding UV (Check Line 53-57). Novelty of the research should be mention at the end of introduction. Recently reported similar work needs more comparison with present work.

The publication suggested by the reviewer (Carbohydrate Polymers, Volume 211, 2019, 181-194) describes a publication where a bionanocomposite is applied for packaging applications. The water contact angle is reported to increased via this treatment. Line 53-57 of our manuscript deals with hydrophilization methods (decreasing water contact angle) for membrane systems. The authors do not understand how the suggested publication might be inserted in this context.

The novelty of the work is already described close to the end of the introduction part, line 84-87.

    The equation 1 should be separated from characterization part. Modify and check it.

Where should the equation be positioned if not in the experimental part? A similar structure (including the positioning of equations in the experimental part) was applied to prior publications by the authors. These publication were successfully published in e.g. Polymers and the positioning of equations was never a problem, thus we would like to keep it at the current position.

    Line 251-253 needs proper references.

The following sentences are written in line 251-253: “A dense top layer is observed in the case of the PES-PA membrane. Due to the polymeric amide network formed by the interfacial polymerization, the pore structure of the PES base membrane is mostly blocked.” These sentences describe the finding of our experiments, here the respective SEM images. The author do not understand why a literature reference should be applied to our own findings.

    Line 276-278, WCA of amide functionalization for membrane should be referred.

The following sentences are written in line 276-278: “The amid functionalization of these membranes resulted in very low WCA values of 37° and 30°, respectively. These membranes are very hydrophilic, while still usable for microfiltration application.” The authors do not understand the comment made by the reviewer.

    Line 310-312 should be rechecked.

The following sentences are written in line 310-312: “The thus fouled membranes were backwashed to remove the loosely bound fouling components, followed by determining the pure water permeance.” The authors do not understand the comment made by the reviewer.

    Conclusion should be technical.

Quantitative data was included in the conclusion.

    No need of the supplementary, you can insert in this part in text. Check and modify it.

The image from the supplementary information was moved to the main article.

Reviewer 3 Report

In this manuscript, “Reduction of Biofouling of a Microfiltration Membrane using Amide Functionalities – Hydrophilization without Changes in Morphology” by Breite et al. reported PES microfiltration membranes were first immobilizing piperazine to the membrane surface via electron beam irradiation and then reacted with 1,3,5-benzenetricarbonyl trichloride. Their permeance, porosity, average pore size, morphology (SEM), chemical composition (XPS), and surface hydrophilicity were investigated. This manuscript is well written, and could be accepted for publication after minor revision. Here are the comments and suggestions:

  1. Some performance of these membranes could be added in the Abstract.
  2. The legend of Fig. 3 could be reversed from PIP-TMC-PIP to REF.
  3. In Fig. 4, the preparation of samples with algae should be described. Algae seem shrink on the surface of membrane, and some biological samples preparation for SEM should be employed.  
  4. In Fig. 5, the symbols could be the same for the same materials in Fig. 3.

Author Response

Thank you very much for considering our manuscript for publication in Polymers. We appreciate your thorough examination of the manuscript, and we carefully revised the manuscript for obtaining now a better quality of the paper. Therefore, we would like to resubmit our manuscript and hope we were able to eliminate all mistakes and misunderstandings according to your suggestions.

We responded to the different review comments as follows:

In this manuscript, “Reduction of Biofouling of a Microfiltration Membrane using Amide Functionalities – Hydrophilization without Changes in Morphology” by Breite et al. reported PES microfiltration membranes were first immobilizing piperazine to the membrane surface via electron beam irradiation and then reacted with 1,3,5-benzenetricarbonyl trichloride. Their permeance, porosity, average pore size, morphology (SEM), chemical composition (XPS), and surface hydrophilicity were investigated. This manuscript is well written, and could be accepted for publication after minor revision. Here are the comments and suggestions:

    Some performance of these membranes could be added in the Abstract.

Some quantitative data was included in the abstract as requested by reviewers 2 and 3. However, due to the limited number of words for the abstract, another sentence was removed.

    The legend of Fig. 3 could be reversed from PIP-TMC-PIP to REF.

The order of the legend of Fig. 3 was reversed: PIP-TMC-PIP (top) to REF (bottom). The same change was applied to Fig. 5 to have the same style for both figures. The symbols of Fig. 5 were adjusted as requested below.

    In Fig. 4, the preparation of samples with algae should be described. Algae seem shrink on the surface of membrane, and some biological samples preparation for SEM should be employed. 

It is true that the algae might shrink because the sputtering of the chromium layer as well as the SEM itself are carried out in vacuum. A respective remark was added to the text (line 354-356). An alternative would be environmental SEM (ESEM). The disadvantage of this method would be the lower resolution. As the authors wanted to make images comparable for membranes with and without fouling, we decided to use the “normal” SEM for both types of samples. The study was focused on the membrane and its modification. The degree of fouling can be depicted by the amount of algae present on the membrane. However, the amount of algae cells will not be altered depending on the SEM method and the state of the algae is of secondary nature.

    In Fig. 5, the symbols could be the same for the same materials in Fig. 3.

The changes were applied as requested.

Round 2

Reviewer 2 Report

Accepted in current form.

Author Response

Many thanks for your support!